# LLM-Oriented Retrieval Tuner

## Abstract

Dense Retrieval (DR) is now considered as a promising tool to enhance the memorization capacity of Large Language Models (LLM) such as GPT3 and GPT-4 by incorporating external memories. However, due to the paradigm discrepancy between text generation of LLM and DR, it is still an open challenge to integrate the retrieval and generation tasks in a shared LLM. In this paper, we propose an efficient **LLM-O**riented **R**etrieval **T**uner, namely LMORT, which decouples DR capacity from base LLM and non-invasively coordinates the optimally aligned and uniform layers of the LLM towards a unified DR space, achieving an efficient and effective DR without tuning the LLM itself. The extensive experiments on six BEIR datasets show that our approach could achieve competitive zero-shot retrieval performance compared to a range of strong DR models while maintaining the generation ability of LLM.

## 1 Introduction

Large language models (LLMs) such as GPT-3 (Brown et al., 2020) and GPT-4 (OpenAI, 2023) have achieved significant success and shown impressive zero/few-shot generalization ability across a wide range of natural language processing tasks (Brown et al., 2020; Kojima et al., 2022). Recently, they are now being functioned as the backbone of autonomous agents consisting of planning, tools, action, and memory, and become a milestone towards Artificial General Agent (AGI) (Weng, 2023).

In a LLM-based autonomous agent, the inclusion of an external memory component, which can aid the agent in retaining and recalling information for a long time (Nematzadeh et al., 2020), holds significant importance. A promising avenue for augmenting LLM's long-term memory lies in dense retrieval (DR), which employs a representation model to map information into dense vectors, and allows efficient identification of relevant information from large-scale vector storage (Karpukhin et al., 2020; Xiong et al., 2021; Ni et al., 2022).

LLM-based retrieval, such as cpt-text (Neelakantan et al., 2022) with large GPT-3 models, presents a promising avenue for establishing external memory. They generally need to fine-tune the LLMs as a retrieval-specific representation models, which is always feasible but suboptimal. Specifically, LLMs maximize the likelihood of the succeeding properly generated token, building upon the history context text (Brown et al., 2020; Touvron et al., 2023). However, the DR task involves transforming existing text into a vector space, whereby text vectors embodying related semantics draw closer, while those representing distinct semantics are distanced further apart (Karpukhin et al., 2020; Wang & Isola, 2020). The divergence paradigm between text generation and DR makes it difficult for LLM-based retriever/agent share a single LLM, resulting in additional model parameters (requiring a retrieval-specific LLM) and longer inference time (i.e., every query needs being re-encoded by the retrieval-LLM). Consequently, achieving compatibility between retrieval and text generation within a unified LLM remains a significant yet largely unresolved problem.

Considering the impressive zero-shot capabilities of LLM across various NLP tasks, we have every reason to support the hypothesis that the original representation of LLM (i.e., the output of a frozen LLM) contains sufficient semantic information for Dense Retrieval (DR), albeit not aligned with the DR space. Inspired by previous works (Wang & Isola, 2020), we introduce the *alignment* and *uniformity*, which represent two important aspects to measure an ideal DR's vector space, to analyze LLM representation space layer by layer. Alignment favors the representation model that assign similar features to similar samples. Uniformity prefers a feature distribution that preserves maximal information for samples. As illustrated in Fig. 1, our analysis reveals that layers of LLM

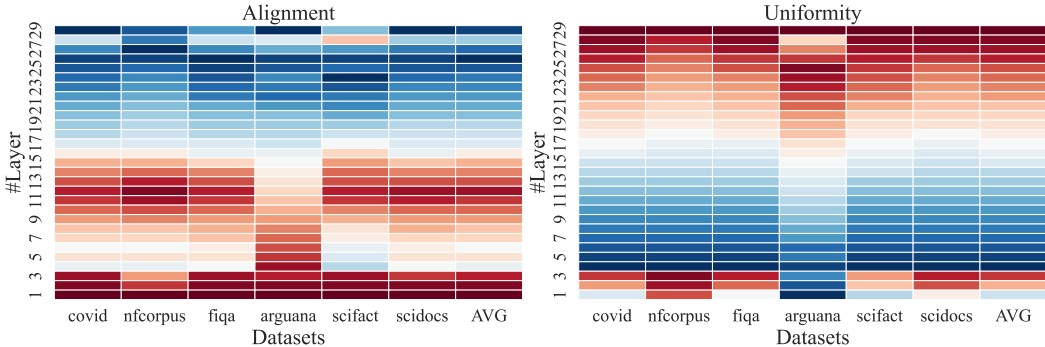

Figure 1: Layer-wise alignment and uniformity analysis in GPT-j-6B. The **redder** the color, the **better** the alignment and uniformity. Conversely, the **bluer** the color, the **worse** alignment and uniformity. The X-axis denotes six BEIR datasets and their average results. The Y-axis represents the layer number of GPT-j-6B (e.g., #1 is the first embedding layer and #29 is the last hidden layer).

with better alignment tend to be less uniform, and conversely, layers with high uniformity exhibit weaker alignment. This observation suggests a more promising direction to solve those problems is to synthesize the *alignment* and *uniformity* of original LLM's representation space, instead of conducting retrieval-specific LLM tuning.

Motivated by the observations mentioned above, we propose an efficient **LLM-O**riented **R**etrieval **T**uner (LMORT), to drives the optimally aligned and uniform layers of the frozen LLM towards a unified DR space. Specifically, the LMORT tuner adapts a Transformer-like structure, comprising two carefully-crafted bidirectional attention operations in each layer. One attention utilizes self-attention to learn features starting from the LLM's optimal alignment (uniformity) layer, and the other cross-attention is operated on LLM's best uniformity (alignment), so as to simultaneously consider uniformity and alignment in a shot. Through the fine-tuning of LMORT with standard DR tasks, the alignment and uniformity properties of the frozen LLM seamlessly merge into a unified space conducive to effective retrieval.

We conduct extensive experiments on six zero-shot retrieval datasets from BEIR benchmark (Thakur et al., 2021), focusing on three LLMs, including GPT2-Large, XL (Radford et al., 2018), and GPT-j-6B (Wang, 2021). LMORT demonstrates significant scalability, with its zero-shot retrieval performance improving by 13% as the size of LLM increases from Large to 6B. Even when compared to strong DR baselines with fine-tuned LLM, tuning just three-layer of LMORT yields competitive performance. Our analysis also indicates that LMORT's effectiveness lies in its resonable utilization of LLM alignment and uniformity, mitigating their original incompatibility. Furthermore, we evaluate LMORT's parameter and training efficiency. After dimensionality reduction, with only a marginal 1% performance decrease, LMORT significantly cuts down training parameters to 2% and training time to 4% compared to LLM-based DR fine-tuning.

## 2 RELATED WORK

Dense Retrieval (DR) based on Pre-trained Language Models (PLMs) entails the process of fine-tuning PLMs into dense representation models (Karpukhin et al., 2020; Ni et al., 2022; Neelakantan et al., 2022). Within this category, masked PLMs with bidirectional attention, such as BERT (Kenton & Toutanova, 2019) and T5 (Raffel et al., 2020), have demonstrated substantial empirical advantages for retrieval tasks. Nevertheless, these benefits tend to diminish in zero-shot DR scenarios, particularly when the PLM is either inadequately sized or not meticulously trained (Thakur et al., 2021).

To enhance the zero-shot generalization capabilities of PLM-based DRs, research communities have explored a variety of training techniques, including the adoption of training data augmentation (Ma et al., 2021), refinement training strategies (Yu et al., 2022). Popular data augmentation techniques include the creation of weakly supervised data through text processes (Lee et al., 2019) such as span corruption (Izacard et al., 2022) and pseudo-query generation (Ma et al., 2021). In parallel, commonly employed training strategies contain retrieval-oriented pre-training (Gao & Callan, 2022; Lu et al., 2021), training-negative iteration (Xiong et al., 2021; Si et al., 2022), and cross-encoder distillation (Ren et al., 2021b; Zhang et al., 2022). In addition to improving training techniques, recent

work has shown that simply increasing the size of T5 to XXL achieves state-of-the-art performance on the zero-shot retrieval benchmark Ni et al. (2022).

Recently, considering the powerful ability of decoder-only large language models (LLMs) on a wide range of NLP tasks Brown et al. (2020); OpenAI (2023), their potential for retrieval tasks has also been explored. Researchers find that simply increasing the size of the LLM can also significantly boost zero-shot retrieval performance (Muennighoff, 2022; Neelakantan et al., 2022). For instance, cpt-text (Neelakantan et al., 2022) fine-tunes the massive 175B GPT-3 Brown et al. (2020) as a dense retriever, achieving superior results on many zero-shot retrieval datasets Thakur et al. (2021). Although directly fine-tuning a larger LLMs is a simpler and effective approach, it comes with a higher training cost and limits the LLM to a retrieval-specific model, making it less compatible with other natural language processing and generation tasks.

In this paper, we employ a distinct approach by fine-tuning a lightweight LLM-oriented retrieval tuner, seamlessly integrated into the LLM without direct alterations to its internal parameters. This approach allows us to unlock the zero-shot retrieval capabilities of the LLM while preserving its versatile generalization abilities at a more cost-effective training expense.

## 3 LLM'S ZERO-SHOT DR CAPABILITY ANALYSIS

In this section, we first recaps the preliminary of Dense Retrieval (DR). Following that, we present the analysis findings regarding zero-shot DR capabilities.

### 3.1 PRELIMINARY OF DR

According to a query $X_q$, DR aims to retrieve a set of relevant passages $X_p^+$ from a large-scale corpus $X_p \in \mathcal{C}$. Specifically, the query $X_q$ and the passage $X_p$ can be encoded as dense representations:

$$\mathbf{x_q} = g(X_q; \phi); \ \mathbf{x_p} = g(X_p; \phi), \tag{1}$$

where $g(\cdot; \phi)$ denotes the representation model with parameters $\phi$. In this way, the whole corpus can be encoded as vector database $\mathcal{V}$ and retained for a long term with a limitless storage capacity.

Then the most related $K$ passages can be retrieved by assessing the similarity between the query vector $\mathbf{x_q}$ and each passage vectors $\mathbf{x_p}$, such as dot product and cosine similarity:

$$\text{Top K}_{\mathbf{x_p} \in \mathcal{V}} \text{sim}(\mathbf{x_q}, \mathbf{x_p}; \phi). \tag{2}$$

### 3.2 ZERO-SHOT DR CAPABILITY ANALYSIS

We carry out an analysis of frozen LLMs from the views of *alignment* and *uniformity* that are two crucial characteristics of an ideal DR space Wang & Isola (2020).

**Layer-wise Dense Representation.** To evaluate the DR potential of LLMs, our initial step involves acquiring a dense representation of the input sequence through the LLM. Achieving this, we transform the hidden states from the LLM's final output layer into dense vectors through mean pooling, and this process is applied to all layers of the LLM.

Given an input sequence $X = \{x_1, ..., x_t, ..., x_n\}$, the LLM processes the sequence $X$ into a set of layered hidden states $\mathbf{H}^l = \{\mathbf{h}_1^l, ..., \mathbf{h}_t^l, ..., \mathbf{h}_n^l\}$, where $1 \leq l < L$ and $L$ is the total layer number of the LLM. Then the intermediate states $\mathbf{H}^l$ of layer $l$ can be pooled into a dense vector $\mathbf{x}^l$:

$$\mathbf{H}^l \leftarrow \text{LLM}(X; \phi_{\text{llm}}),$$
$$\mathbf{x}^l = f(\mathbf{H}^l), \tag{3}$$

where $\phi_{\text{llm}}$ and $f$ denotes the LLM's parameters and the mean pooling operation, respectively. Once the layer-wise dense representation is established, the alignment and uniformity of the representation space at each LLM layer can be evaluated.

**Layer-wise Alignment Analysis.** Alignment requires that two samples forming a positive pair should be mapped to nearby features in the representation space, i.e., the distance between positive samples is excepted to be closer. Formally, let be a positive pair $(X, X^+) \sim p_{\text{pos}}$, which is

processed into LLM's layered hidden states $\mathbf{H}_X^l$ and $\mathbf{H}_{X^+}^l$ (Eq. 3), where $0 \leq l < L$. Then each layer of hidden states $\mathbf{H}_{(\cdot)}^l$ is then mapped into a representation vector $f(\cdot)$ through pooling and normalization operations. The alignment loss $\mathcal{L}_{\text{align}}$ is introduced to measure the expected distance between positive pairs:

$$\mathcal{L}_{\text{align}}(f) \triangleq \mathbb{E}_{(X, X^+) \sim p_{\text{pos}}} || f(\mathbf{H}_X^l) - f(\mathbf{H}_{X^+}^l) ||_2^2. \tag{4}$$

**Layer-wise Uniformity Analysis.** Uniformity prefers representation vectors should be uniformly distributed on the normalized feature space, preserving as much information of the sample as possible. Similar to alignment analysis, we measure the uniformity loss of each LLM layer. The uniformity loss $\mathcal{L}_{\text{uniform}}$ is defined as the expected pairwise potential of all sample pairs $(X, Y) \overset{\text{i.i.d}}{\sim} p_{\text{data}}$:

$$\mathcal{L}_{\text{uniform}}(f) \triangleq \log \mathbb{E}_{(X, Y) \overset{\text{i.i.d}}{\sim} p_{\text{data}}} e^{-2||f(\mathbf{H}_X^l) - f(\mathbf{H}_Y^l)||_2^2}. \tag{5}$$

**Layer-wise Analysis Results.** We conduct the layer-wise alignment and uniformity analysis on three causal LLMs of different sizes: GPT2-Large (0.75B, 37 layers), GPT2-XL (1.5B, 49 layers) and GPT-j-6B (6B, 29 layers). The analysis data are six zero-shot retrieval datasets from the BEIR benchmark Thakur et al. (2021) [1]: TREC-COVID, NFCorpus, FiQA, ArguAna, SciFact, and SCI-DOCS. In these datasets, the query and the relevant passage are regarded as positive pairs $(X, X^+)$ in the alignment analysis. When evaluating uniformity, the pairwise pair $(X, Y)$ can be uniformly sampled from the query set and the passage set.

Figure 1 illustrates the alignment and uniformity losses computed from the representation space of each layer of GPT-j-6B on these datasets (more results are shown in Appendix C). The results suggest that the inherent nature of LLMs makes it challenging to simultaneously optimize alignment and uniformity within a single layer, as these two properties tend to be mutually exclusive. This observation becomes more pronounced for larger LLMs. Additionally, we also observe that lower layers exhibit better alignment, while higher layers tend to excel in uniformity. This finding is consistent with previous research Sajjad et al. (2022): the LLM captures shallow concepts at the low layers, such as lexical n-grams. Meanwhile, the lexical overlap is an important feature of positive pairs (alignment); while top layers capture richer and more abstract information, such as morphology, semantics and syntax, revealing that higher layers preserve more information (uniformity).

Through the above analysis, we conclude that the representation spaces of frozen LLMs possess the alignment and uniformity characteristics necessary for an effective retrieval space. However, these two properties are distributed across different layers of the LLM. Consequently, these observations inspire the idea presented in Section 4, which aims to unleash the zero-shot DR capability of LLM, by tuning the alignment and uniformity of the LLM into a unified output representation space.

## 4 LLM-ORIENTED RETRIEVAL TUNER (LMORT)

Motivated by the insights discussed in Section 3, we intuitively propose a LLM-oriented retrieval tuner, namely LMORT, which non-invasively tunes the optimal alignment and uniformity layer of LLMs into a unified representation space to achieve a effective LLM-oriented retrieval.

### 4.1 LMORT'S ARCHITECTURE

Figure 2 illustrates the architecture of LMORT, which is a multi-layer architecture built on top of LLM's optimal alignment and uniformity layers. Each LMORT layer contains three carefully-designed sub-layers. Next, we first introduce the selection of LLM's alignment and uniformity layers and then describe the details of LMORT.

**LLM's Align & Uniform Layers.** As per the analysis method introduced in Section 3, we select the alignment layer with the lowest alignment loss (Eq. 4) and the uniform layer with the lowest uniformity loss (Eq. 5). Specifically, given an input sequence $X = \{x_1, ..., x_t, ..., x_n\}$, LLM processes

---
[1]Due to the high cost of LLM-oriented evaluation, we have chosen six reasonably sized datasets for experiments from all 18 BEIR datasets. More details of these datasets can be found in Appendix B.

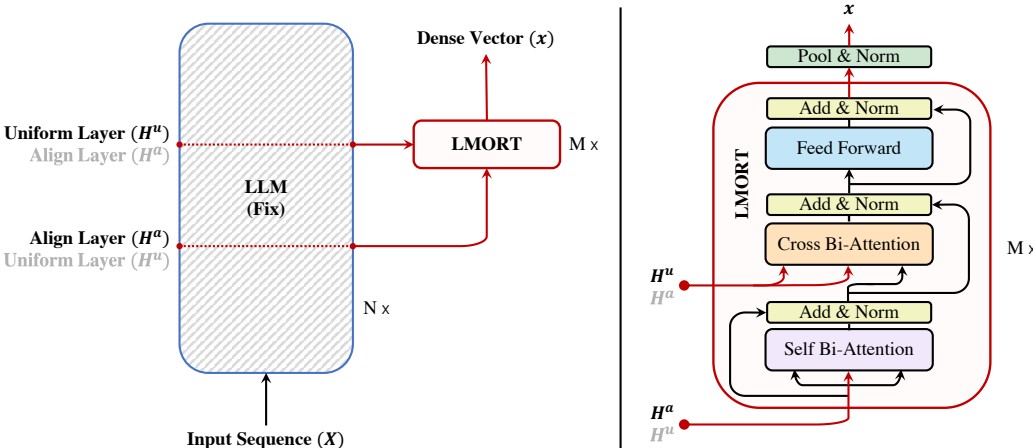

Figure 2: Illustration of LLM-Oriented Retrieval Tuner (LMORT). The total layer number of LMORT is much less than that of the frozen LLM ($M \ll N$).

the sequence $X$ into a set of layered hidden states $\mathbf{H}^l = \{\mathbf{h}_1^l, ..., \mathbf{h}_t^l, ..., \mathbf{h}_n^l\}$, where $0 \leq l < L$. the LLM's align layer and uniform layer are denoted as $\mathbf{H}^a$ and $\mathbf{H}^u$, respectively:

$$\mathbf{H}^a, \mathbf{H}^u \leftarrow \text{LLM}(X; \phi_{\text{llm}}), \tag{6}$$

**LMORT's Layer-wise Structure.** Each LMORT block consists of two carefully-designed bidirectional attention sub-layers (i.e., *self bi-attention* and *cross bi-attention* sub-layers) and one feedforward layer. Referring to vanilla Transformer blocks, residual connections surround each sublayer, and layer normalization follows. The two attention sub-layers are directed towards the LLM's align layer $\mathbf{H}^a$ and uniform layer $\mathbf{H}^u$, respectively:

- *Self Bi-Attention*. The first attention sub-layer utilizes a bi-directional attention operation, whereby the attention matrices $\mathbf{Q}^a$, $\mathbf{K}^a$, and $\mathbf{V}^a$ are all mapped from the LLM's align layer $\mathbf{H}^a$ and each token at a given position interacts with tokens from all positions in the sequence. This attention mechanism facilitates the identification and capturing contextual features of sequence from LLM's alignment perspective:

$$\text{Self-Attention}(\mathbf{Q}^a, \mathbf{K}^a, \mathbf{V}^a) = \text{softmax}\left(\frac{\mathbf{Q}^a(\mathbf{K}^a)^T}{\sqrt{d_k}}\right)\mathbf{V}^a. \tag{7}$$

- *Cross Bi-Attention*. The second attention sub-layer also employs bi-directional attention, but with a significant difference: The key $\mathbf{K}^u$ and value $\mathbf{V}^u$ is sourced from the LLM's uniform layer's output $\mathbf{H}^u$, while query $\mathbf{Q}^s$ are obtained from the previous *Self Bi-Attention*'s output. This design establishes an inner connection within LLM's align and uniform layers, enabling LMORT to narrow the gap between them:

$$\text{Cross-Attention}(\mathbf{Q}^s, \mathbf{K}^u, \mathbf{V}^u) = \text{softmax}\left(\frac{\mathbf{Q}^s(\mathbf{K}^u)^T}{\sqrt{d_k}}\right)\mathbf{V}^u. \tag{8}$$

It should be noted that the connections between LMORT's two attention sub-layers and LLM's alignment and uniformity layers can be inter-changed. For instance, Self-Attention$(\mathbf{Q}^u, \mathbf{K}^u, \mathbf{V}^u)$ can also be applied to the uniform layer $\mathbf{H}^u$, while Cross-Attention$(\mathbf{Q}^s, \mathbf{K}^a, \mathbf{V}^a)$ can be directed towards the alignment layer $\mathbf{H}^a$. This connection mode is regarded as one of the hyper-parameters.

Lastly, LMORT's output representation $\mathbf{H}^o$ is converted into a dense vector $\mathbf{x}$ through the mean pooling operation $f(\cdot)$:

$$\begin{aligned} \mathbf{H}^o &= \text{LMORT}(\mathbf{H}^a, \mathbf{H}^u; \theta), \\ \mathbf{x} &= f(\mathbf{H}^o), \end{aligned} \tag{9}$$

where $\theta$ represents the parameters of the LMORT. In this way, large-scale input sequences $\mathcal{X}$ can be encoded and stored as dense vectors $\mathcal{V}$.

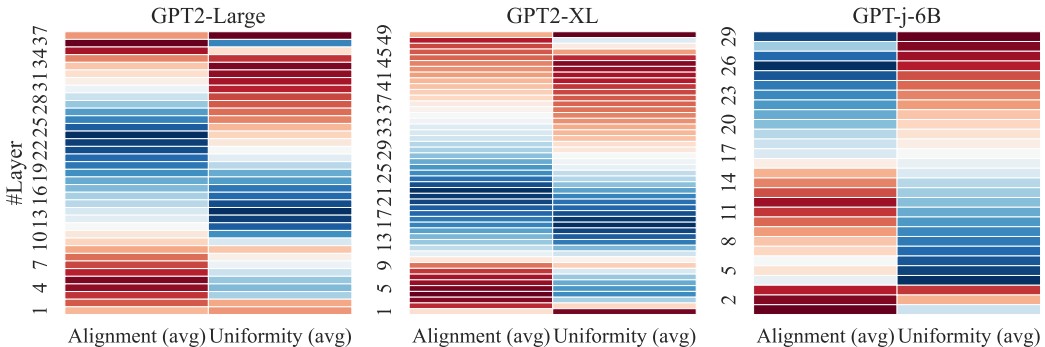

Figure 3: The average results of layer-wise alignment and uniformity estimation on six BEIR datasets. The **redder** the color, the **better** the alignment and uniformity. Conversely, the **bluer** the color, the **worse** alignment and uniformity. The Y-axis represents the layer number of three GPTs.

## 4.2 LMORT'S TRAINING

We employ the standard DR training method to fine-tune LMORT. Formally, let be a training query set $\mathcal{Q}$ and a corpus $\mathcal{C}$, where each query $X_q$ is labeled with a set of relevant passages $X_p^+ \in \mathcal{C}_{X_q}^+$, and negative passages $\mathcal{C}_{X_q}^-$ sampled from the rest corpus $\mathcal{C} \setminus \mathcal{C}_{X_q}^+$. The learning objective can be formulated as optimizing parameters $\theta$ of LMORT, in such a way, positive pairs of the query and positive passages $(X_q, X_p^+)$ have higher similarity than the negative ones $(X_q, X_p^-)$:

$$\theta^* = \arg\min_\theta \sum_{X_q \in \mathcal{Q}} \sum_{X_p^+ \in \mathcal{C}_{X_q}^+} -\log \frac{\exp\left(\text{sim}(\mathbf{x_q}, \mathbf{x_p^+}; \theta, \widetilde{\phi}_{\text{llm}})\right)}{\exp\left(\text{sim}(\mathbf{x_q}, \mathbf{x_p^+}; \theta; \widetilde{\phi}_{\text{llm}})\right) + \sum_{X_p^- \in \mathcal{C}_{X_q}^-} \exp\left(\text{sim}(\mathbf{x_q}, \mathbf{x_p^-}; \theta; \widetilde{\phi}_{\text{llm}})\right)},$$
(10)

where the dense representation of query $\mathbf{x_q}$ and passage $\mathbf{x_p}^{+/-}$ are encoded through LLM (Eq. 6) and LMORT (Eq. 9). During training, we only tune the LMORT's parameters ($\theta$) and freeze the all parameters of the LLM ($\widetilde{\phi}_{\text{llm}}$), where the optimization gradient is not passed back to the LLM.

## 5 EXPERIMENTS

In this section, we begin by detailing the experimental setups, followed by conducting hyperparameter studies, comparisons, and ablation experiments on LMORT. Additionally, we perform an in-depth analysis of alignment, uniformity, efficiency, and scalability.

### 5.1 EXPERIMENTAL SETUPS

**Evaluation Datasets.** We employ six zero-shot retrieval datasets, the same analysis data in Section 3, as testing data. These datasets include TREC-COVID (Voorhees et al., 2021), NFCorpus (Boteva et al., 2016), FiQA (Maia et al., 2018), ArguAna (Wachsmuth et al., 2018), SciFact (Wadden et al., 2020), and SCIDOCS (Cohan et al., 2020). We evaluate model performance using the standard NDCG@10 metric. For training, we leverage MS MARCO (Nguyen et al., 2016), which annotates about 500k web query-positive passages, and use training negatives released by sentence-transformers (Reimers & Gurevych, 2019). More details presents in Appendix A and B.

**LMORT Implementation.** We select GPTs as the target LLMs, specifically containing GPT2-Large (0.75B), GPT2-XL (1.5B) and GPT-j-6B. For all our training runs, we maintain a batch size of 8, a learning rate of 5e-6, and train for a total of 3 epochs. We evaluate the models using the checkpoint from the last training step, without selecting checkpoints based on testing performance. We employ a single RTX 3090 GPU (24GB) for GPT2-Large and XL, while GPT-j-6B utilizes four A100 GPUs (40GB). More implementation details are listed in Appendix A.

## 5.2 HYPER-PARAMETER STUDY

We explore three hyperparameters of LMORT: *(1)* Connection Mode: This pertains to how LLM's optimal alignment (A) and uniformity (U) layers are connected within LMORT. We test two connection methods: (A→U) self-attention to A and cross-attention to U; (U→A) self-attention to U and cross-attention to A. *(2)* Number of LMORT Layers: We determine the total number of LMORT layers to be utilized. *(3)* Performance-scaling with base-LLM size: We assess performance scalability under various base-LLMs with different parameter sizes.

**Connection Mode.** The optimal A&U layer of LLMs is determined by the alignment and uniformity loss (Eq. 4 and Eq. 5). Therefore, we firstly assess these two losses for each layer of GPT2-Large, XL, and GPT-j-6B across six BEIR datasets. The resulting average losses are visualized in Fig. 3. We observe that the A layers differ among the three LLMs, but their U layers are consistently the final layers. Tab. 1 shows the results of specific layer selection and connection mode.

Table 1: LLM's optimal alignment (A) and uniformity (U) layers and the average NDCG@10 scores of LMORT on six BEIR datasets. A→U denotes LMORT applies self-attention on A layer and cross-attention to U layer. U→A means using self-attention on U and cross-attention to A. worst A&U denotes connecting LMORT to the worst A&U layers.

| LLMs | A **layer** | U **layer** | LMORT (NDCG@10) | | |
| --- | --- | --- | --- | --- | --- |
| | | | A→U | U→A | worst A&U |
| GPT2-Large | #36 | #37 | **0.296** | 0.294 | 0.248 |
| GPT2-XL | #4 | #49 | 0.342 | **0.355** | 0.167 |
| GPT-j-6B | #1 | #29 | **0.425** | 0.417 | 0.324 |

As shown in Tab. 1, Large and 6B prefer the A→U connection, whereas XL is better suited for the U→A connection within LMORT. When LMORT is connected to the LLM layers with the worst alignment and uniformity, its retrieval performance significantly declines across all three LLM scenarios. This reveals the critical role of selecting and connecting LLM's A&U layers for LMORT's effectiveness.

**Number of LMORT Layers.** We further conduct experiments with these LLMs to investigate how the number of LMORT layers affects their zero-shot retrieval performance. The results exhibited in Fig. 4 reveals that various LLMs show effective retrieval performance with a very few number of LMORT layers. Notably, larger LLMs require fewer LMORT layers, i.e., the optimal LMORT layer count is seven for GPT2-Large ($L$=7), five for GPT2-XL ($L$=5), and just three for GPT-j-6B ($L$=3).

**Performance-scaling with base-LLM size.** As LMORT transitions from GPT2-Large to XL and then GPT-j-6B, its retrieval performance consistently sees improvements of 6% and 7%, respectively. This significant performance-scaling capability highlight the effectiveness of the lightweight LMORT in unlocking the retrieval potential of LLMs without necessitating any tuning of the LLM.

## 5.3 COMPARISON & ABLATION STUDY

**Baselines.** We experiment four different ablated versions of LMORT. First, *(1)* we shift LMORT from its best align and uniform layers to the worst A&U layers of LLMs. *(2&3)* Then, we remove LMORT's cross-bi-attention, only retaining the self-bi-attention to A or U. Since the the last layer of three LLMs consistently serves as U layers, self-attention to U is equivalent to applying self-attention on top of LLMs. *(4)* To provide a basis for comparison, we also test the effectiveness of applying self-attention to the embedding layer (E) of LLM, indicating LLMs have not been used. Notably, in GPT-j-6B, layer A is identical to layer E.

Apart from the ablated baselines, we also present results from four publicly classic baselines for comparison: BM25 (Yang et al., 2017), DPR (Karpukhin et al., 2020), GTR-XXL (Ni et al., 2022), and cpt-text-L (Neelakantan et al., 2022). BM25 is a sparse retriever, demonstrating strong performance in zero-shot retrieval tasks. On the other hand, DPR, GTR-XXL, and cpt-text-L are dense retrievers employing BERT-base, T5-XXL-encoder (4.5B), and GPT3-6B as their fine-tuning backbones, respectively. It is worth noting that the proposal of LMORT is to equip LLM with zero-shot retrieval capability without altering any internal states of LLMs, instead of achieving state-of-the-art on BEIR. Hence, we do not reference methods that achieve SOTA results through more sophisticated training techniques, even though these skills could theoretically be applied to LMORT.

**Evaluation results.** Tab. 2 presents the overall results. Among three different size of base-LLM settings, the performance of the four-ablated LMORT versions is significantly worse compared to the full version. Specifically, when LMORT is converted to the worst A&U layer of LLMs, retrieval

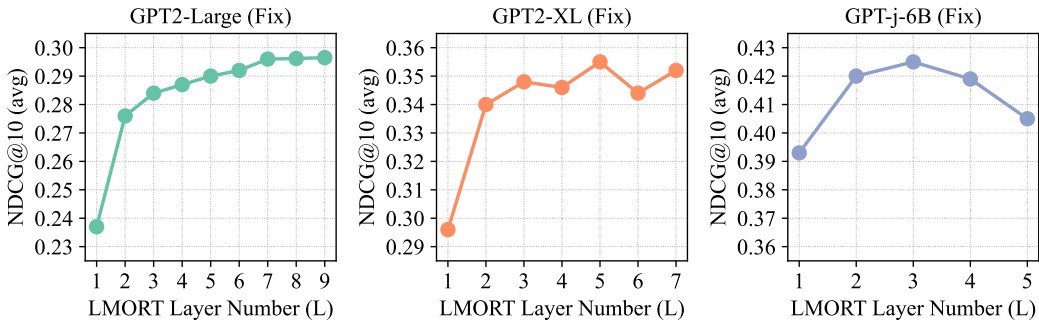

Figure 4: The average NDCG@10 results of LMORT with different layer number on three LLMs (GPT2-Large, GPT2-XL, GPT-j-6B). The X-axis means the total layer number of LMORT. The Y-axis denotes the average NDCG@10 scores of six BEIR datasets.

Table 2: Overall NDCG@10 results on six BEIR datasets. L means the layer number of LMORT. `A`, `U`, and `E` denotes the alignment (`A`), uniformity (`U`), and embedding (`E`) layers of LLMs, respectively.

| LLMs | Methods | COVID | NFCorpus | FiQA | ArguAna | SciFact | SCIDOCS | AVG |
|---|---|---|---|---|---|---|---|---|
| GPT2-Large (Fix) | **LMORT (L=7)** | 0.455 | **0.224** | 0.164 | **0.393** | 0.451 | **0.091** | **0.296** |
| | self/cross-attn to worst `A&U` | 0.405 | 0.189 | **0.176** | 0.235 | 0.396 | 0.085 | 0.248 |
| | only self-attn to `A` | 0.371 | 0.206 | 0.143 | 0.349 | 0.448 | 0.076 | 0.266 |
| | only self-attn to `U` (top LLM) | 0.373 | 0.203 | 0.131 | 0.376 | **0.471** | 0.076 | 0.272 |
| | only self-attn to `E` (w/o LLM) | **0.481** | 0.175 | 0.134 | 0.316 | 0.368 | 0.090 | 0.261 |
| GPT2-XL (Fix) | **LMORT (L=5)** | **0.600** | **0.236** | **0.219** | **0.428** | **0.526** | **0.122** | **0.355** |
| | self/cross-attn to worst `A&U` | 0.287 | 0.156 | 0.114 | 0.094 | 0.281 | 0.068 | 0.167 |
| | only self-attn to `A` | 0.499 | 0.207 | 0.165 | 0.330 | 0.524 | 0.107 | 0.305 |
| | only self-attn to `U` (top LLM) | 0.418 | 0.223 | 0.168 | 0.426 | 0.449 | 0.088 | 0.295 |
| | only self-attn to `E` (w/o LLM) | 0.472 | 0.171 | 0.123 | 0.321 | 0.469 | 0.087 | 0.274 |
| GPT-j-6B (Fix) | **LMORT (L=3)** | **0.735** | 0.280 | **0.251** | **0.476** | **0.679** | **0.126** | **0.425** |
| | self/cross-attn to worst `A&U` | 0.698 | 0.286 | 0.229 | 0.405 | 0.205 | 0.122 | 0.324 |
| | only self-attn to `A` (w/o LLM) | 0.516 | 0.193 | 0.144 | 0.308 | 0.503 | 0.094 | 0.293 |
| | only self-attn to `U` (top LLM) | 0.707 | **0.290** | 0.248 | 0.432 | 0.646 | 0.115 | 0.406 |
| | only self-attn to `E` (w/o LLM) | 0.516 | 0.193 | 0.144 | 0.308 | 0.503 | 0.094 | 0.293 |
| *For Reference: Sparse Retrieval and Dense Retrieval (fine-tuning all parameters)* | | | | | | | | |
| | BM25 | **0.656** | 0.325 | 0.236 | 0.315 | 0.665 | 0.158 | 0.393 |
| | DPR (BERT-base) | 0.588 | 0.234 | 0.206 | 0.394 | 0.494 | 0.119 | 0.339 |
| | GTR-XXL (T5-enc-4.5B) | 0.501 | 0.342 | **0.467** | **0.540** | 0.662 | **0.161** | **0.445** |
| | cpt-text-L (GPT3-6B) | 0.562 | **0.380** | 0.452 | 0.469 | **0.744** | n.a. | n.a. |

performance degrades by 5%, 19%, and 10% for GPT2-Large, XL, and GPT-j-6B, respectively. On the other hand, removing cross-bi-attention from LMORT and retaining only self-bi-attention leads to the largest drops in performance, with decreases of 3%, 6%, and 13% for Large, XL, and 6B, respectively. These results underscore the critical importance of selecting the `A&U` layer of LLMs and their connections to the self/cross-attention of LMORT.

Compared to strong sparse and dense retrievers on BEIR, the outcome of LMORT is quite promising, considering that the base LLM remains entirely frozen, with only 3 plugin LMORT layers fine-tuned. LMORT initially falls behind when mounted on GPT2-Large. However, its performance greatly improves when transitioning to GPT2-XL, surpassing DPR. Furthermore, with the base LLM scaled up to 6B size, LMORT outperforms BM25 and DPR by 3% and 9%, respectively, trailing behind GTR-XXL by just 2%. As we known, GTR-XXL and cpt-text-L leverage additional training data, while LMORT trains solely with MARCO. We thus have a reasonable expectation that LMORT's performance can be further enhanced through the utilization of data augmentation techniques.

## 5.4 FURTHER ANALYSIS

In this sub-section, we further analyze LMORT's alignment and uniformity, and then quantify the parameter and training efficiency of LMORT.

**LMORT's alignment & uniformity.** We analyze alignment and uniformity losses in the output (`O`) layer of LMORT (GPT-j-6B) and compare them to the optimal align (`A`) and uniform (`U`) layer of the LLM. These results shown in Fig. 5 highlight LMORT's ability to achieve a better balance between alignment and uniformity within the same representation space. However, this balance does

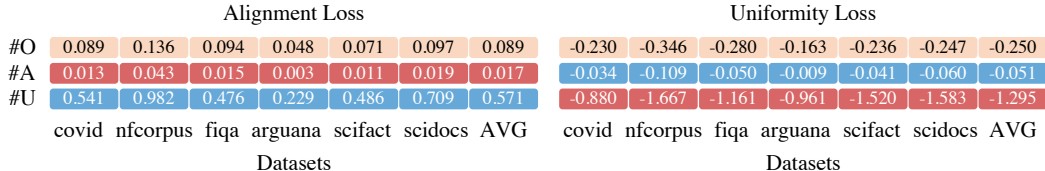

Figure 5: The alignment and uniformity analysis of LMORT (GPT-j-6B) on six BEIR datasets. `#O` means the output layer of LMORT. `#A` and `#U` denotes the optimal alignment and uniformity layer of the LLM, respectively. The **minimum** the loss, the **better** the alignment and uniformity. Conversely, the **maximum** the loss, the **worse** alignment and uniformity.

Table 3: NDCG@10 results of the standard and dimension-reduced versions of LMORT (L=3), where LLM is GPT-j-6B. #D denotes the dimension size. #P denotes the ratio of parameters of LMORT to the base LLM. #T represents the ratio of the average time of each training step to that of directly fine-tuning the LLM.

| LMORT | #D | #P | #T | COVID | NFCorpus | FiQA | ArguAna | SciFact | SCIDOCS | AVG |
|---|---|---|---|---|---|---|---|---|---|---|
| standard | 4096 | 13% | 14% | **0.735** | **0.280** | **0.251** | **0.476** | **0.679** | **0.126** | **0.425** |
| dim-reduced | 1024 | 2% | 4% | 0.734 | **0.280** | 0.248 | 0.427 | 0.663 | 0.122 | 0.412 |

come at some cost to optimal alignment and uniformity. How to simultaneously maintain/surpass the optimal `A` and `U` provided by LLM in a unified space, is a potential avenue for future research.

**LMORT's parameter efficiency.** Additionally, we conduct an assessment of LMORT's parameter size. Tab. 3 presents the results of the standard LMORT mounted on GPT-j-6B, which shares the same hidden vector dimensions as GPT-j-6B, and the dimension-reduced version of LMORT. The standard LMORT comprises only 13% of LLM's parameters, while the dimension-reduced version contains a mere 2% of LLM's parameters, with just a 1% drop in retrieval performance.

**LMORT's training efficiency.** We also analyze the training efficiency of LMORT. Specifically, we compare the cost time per training step between training LMORT and fine-tuning LLM on a single A100 GPU, where we set the batch size and number of positive and negative passages to one, and the input length to 32. The results of Tab. 3 show that standard LMORT only requires 14% of the time of each training step of directly fine-tuning LLM, and LMORT after dimensionality reduction (Appendix A), even reduces the training time to 4% of that of fine-tuning LLM. Such high training efficiency is due to the mechanism that LMORT avoids propagating gradients back to LLM.

## 6    LIMITATIONS AND FUTURE WORK

**Limitations.** LMORT still lags behind the retrieval performance achieved through LLM-based fine-tuning at the same scale. We believe this performance gap will narrow with the size of base-LLM. Moreover, it's important to emphasize that LMORT can only be used with open-source LLMs because it necessitates access to the LLM's hidden state.

**Future work.** LMORT offers a obvious advantage in its compatibility with LLM's retrieval and generation abilities. This makes it an suitable choice for memory-enhanced generation scene, e.g., dealing with long-text modeling and long-range conversations. LMORT can effectively store LLM-processed information for long periods, facilitating quick retrieval when necessary. These retrieved memories can be seamlessly integrated into the latest modeling sequence, ensuring consistent long-range modeling. The application of LMORT will be left for future research.

## 7    CONCLUSION

In this paper, we initially conduct a layer-wise analysis on the representation space of the frozen LLM from the perspective of alignment and uniformity traits for DR, observing mutually exclusive nature between those two metrics. Subsequently, we further propose a novel tuner, namely LMORT, which strikes a trade-off between the optimal alignment and uniformity layers of LLM, establishing an effective dense representation for DR. Extensive experiments on six BEIR datasets show that LMORT could unlock zero-shot capacity of LLM and achieve competitive performance in terms of retrieval ability and parameter/training efficiency. The plugin paragram of LMORT could unify the DR and text generation in a shared LLM, providing a new alternative for memory-augmented LLM.

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

## A  TRAINING DETAILS

**Training dataset.** We employ the MS MARCO passage ranking dataset (Nguyen et al., 2016) as the training data, which is curated from actual user search queries originating from the Bing search engine. Within MARCO, there are approximately 500k pairs of training queries and corresponding positive passages. The training negatives used are sampled from the collected negatives provided by sentence-transformer [2]. We have made the training negatives available in supplementary material.

Table 4: Hyper-parameter of training LMORT.

| Hyperparameters | GPT2-Large (Fix) | GPT2-XL (Fix) | GPT-j-6B (Fix) |
|---|---|---|---|
| LMORT layer number | 7 | 5 | 3 |
| Batch size (query size) | 8 | 8 | 8 |
| Positive size per query | 1 | 1 | 1 |
| Negative size per query | 7 | 7 | 15 |
| Max query length | 32 | 32 | 32 |
| Max passage length | 128 | 128 | 128 |
| Learning rate | 5e-6 | 5e-6 | 5e-6 |
| Optimizer | AdamW | AdamW | AdamW |
| Scheduler | WarmupDecayLR | WarmupDecayLR | WarmupDecayLR |
| Warmup ratio | 0.1 | 0.1 | 0.1 |
| Training epoch | 3 | 3 | 3 |
| FP16 | ✓ | ✓ | ✓ |
| DeepSpeed | ✗ | ✗ | ZeRO-2 |

**Training hyper-parameter.** We outline all of our training hyperparameters in Tab. 4. During training LMORT (GPT-j-6B), we use DeepSpeed ZeRO-2 technology (Ren et al., 2021a) for gradient partitioning. Additionally, the corresponding code can also be found in the supplementary material.

**Dimensionality reduction.** In Sec 5.4, we simply employ two-layer MLPs to perform dimensionality reduction on LMORT (GPT-j-6B). The 4096-dimensional hidden states from the LLM's alignment and uniformity layer passes through two MLP layers with sizes of 8192 and 1024 dimensions, respectively. This results in a 4-fold reduction in dimensionality, reducing the inner dimension size of LMORT to 1024. Further implementation details can be located in the supplementary material.

## B  EVALUATION DETAILS

**Evaluation datasets.** Six BEIR datasets (Thakur et al., 2021) are used for zero-shot DR evaluation, i.e., TREC-COVID (Voorhees et al., 2021), NFCorpus (Boteva et al., 2016), FiQA-2018 (Maia et al., 2018), ArguAna (Wachsmuth et al., 2018), SciFact (Wadden et al., 2020), and SCIDOCS (Cohan et al., 2020). Tab. 5 shows their statistics. The Normalised Cumulative Discount Gain (NDCG@10) score on the test set is used as the metric, which is consistent with prior work (Thakur et al., 2021).

Table 5: Statistics of evaluation datasets.

| Dataset | Test Query | Corpus |
|---|---|---|
| TREC-COVID | 50 | 171332 |
| NFCorpus | 323 | 3633 |
| FiQA-2018 | 648 | 57638 |
| ArguAna | 1406 | 8674 |
| SciFact | 300 | 5183 |
| SCIDOCS | 1000 | 25657 |

**Evaluation hyper-parameters.** We keep the same evaluation hyperparameter settings as utilized in previous research (Yu et al., 2022). Detailed hyperparameters can be found in its public evaluation script [3]. During the evaluation of LMORT (GPT-j-6B), we adopt the DeepSpeed Zero-3 technology (Ren et al., 2021a) for parameter partitioning.

---

[2] https://huggingface.co/datasets/sentence-transformers/msmarco-hard-negatives

[3] https://github.com/OpenMatch/OpenMatch/tree/master/scripts/BEIR

## C  DETAILED LAYER-WISE ANALYSIS RESULTS

Figures 6, 7, and 8 show in detail the layer-wise analysis results on the alignment loss and uniformity loss for GPT2-Large (37 layers), GPT2-XL (49 layers), and GPT-j-6B (29 layers), as described in Section 3. Lower loss values in these figures indicate a higher level of alignment and uniformity.

| #Layer | GPT2-Large (Alignment Loss) | | | | | | | #Layer | GPT2-Large (Uniformity Loss) | | | | | | |
|---|---|---|---|---|---|---|---|---|---|---|---|---|---|---|---|
| | COVID | NFCorpus | FiQA | ArguAna | SciFact | SCIDOCS | AVG | | COVID | NFCorpus | FiQA | ArguAna | SciFact | SCIDOCS | AVG |
| 37 | 0.770 | 1.175 | 0.779 | 0.354 | 0.568 | 0.929 | 0.763 | 37 | -1.219 | -1.546 | -1.576 | -1.518 | -1.381 | -1.391 | -1.439 |
| 36 | 0.114 | 0.214 | 0.106 | 0.045 | 0.100 | 0.144 | 0.121 | 36 | -0.156 | -0.293 | -0.226 | -0.210 | -0.235 | -0.249 | -0.228 |
| 35 | 0.166 | 0.824 | 0.168 | 0.055 | 0.146 | 0.223 | 0.264 | 35 | -0.241 | -0.922 | -0.379 | -0.261 | -0.358 | -0.409 | -0.428 |
| 34 | 0.359 | 2.377 | 0.483 | 0.079 | 0.330 | 0.601 | 0.705 | 34 | -0.562 | -1.334 | -0.979 | -0.374 | -0.777 | -1.022 | -0.841 |
| 33 | 0.793 | 2.863 | 1.087 | 0.104 | 0.699 | 1.237 | 1.131 | 33 | -1.027 | -0.741 | -1.501 | -0.489 | -1.248 | -1.481 | -1.081 |
| 32 | 1.116 | 2.967 | 1.457 | 0.116 | 1.009 | 1.607 | 1.379 | 32 | -1.147 | -0.495 | -1.468 | -0.532 | -1.388 | -1.404 | -1.072 |
| 31 | 1.327 | 2.994 | 1.686 | 0.125 | 1.190 | 1.810 | 1.522 | 31 | -1.111 | -0.387 | -1.326 | -0.544 | -1.364 | -1.254 | -0.998 |
| 30 | 1.465 | 2.990 | 1.816 | 0.132 | 1.335 | 1.934 | 1.612 | 30 | -1.034 | -0.323 | -1.181 | -0.561 | -1.300 | -1.115 | -0.919 |
| 29 | 1.605 | 2.995 | 1.913 | 0.136 | 1.470 | 2.043 | 1.694 | 29 | -0.936 | -0.270 | -1.048 | -0.553 | -1.225 | -0.982 | -0.836 |
| 28 | 1.708 | 2.979 | 1.985 | 0.141 | 1.585 | 2.111 | 1.752 | 28 | -0.825 | -0.229 | -0.901 | -0.563 | -1.149 | -0.858 | -0.754 |
| 27 | 1.764 | 2.957 | 2.043 | 0.139 | 1.674 | 2.158 | 1.789 | 27 | -0.719 | -0.196 | -0.772 | -0.537 | -1.039 | -0.739 | -0.667 |
| 26 | 1.817 | 2.943 | 2.083 | 0.137 | 1.757 | 2.196 | 1.822 | 26 | -0.635 | -0.174 | -0.667 | -0.501 | -0.951 | -0.649 | -0.596 |
| 25 | 1.866 | 2.927 | 2.118 | 0.136 | 1.844 | 2.225 | 1.853 | 25 | -0.542 | -0.152 | -0.569 | -0.475 | -0.845 | -0.559 | -0.524 |
| 24 | 1.900 | 2.913 | 2.134 | 0.132 | 1.926 | 2.235 | 1.873 | 24 | -0.477 | -0.135 | -0.495 | -0.441 | -0.763 | -0.496 | -0.468 |
| 23 | 1.898 | 2.877 | 2.107 | 0.131 | 1.983 | 2.218 | 1.869 | 23 | -0.428 | -0.123 | -0.441 | -0.423 | -0.697 | -0.442 | -0.426 |
| 22 | 1.898 | 2.834 | 2.081 | 0.127 | 2.018 | 2.199 | 1.860 | 22 | -0.379 | -0.111 | -0.386 | -0.399 | -0.636 | -0.392 | -0.384 |
| 21 | 1.884 | 2.778 | 2.042 | 0.125 | 2.038 | 2.163 | 1.838 | 21 | -0.338 | -0.101 | -0.340 | -0.386 | -0.575 | -0.350 | -0.348 |
| 20 | 1.870 | 2.740 | 2.013 | 0.121 | 2.071 | 2.137 | 1.825 | 20 | -0.306 | -0.093 | -0.304 | -0.359 | -0.522 | -0.319 | -0.317 |
| 19 | 1.851 | 2.681 | 1.973 | 0.114 | 2.092 | 2.103 | 1.802 | 19 | -0.263 | -0.084 | -0.260 | -0.329 | -0.464 | -0.284 | -0.281 |
| 18 | 1.824 | 2.636 | 1.935 | 0.101 | 2.089 | 2.065 | 1.775 | 18 | -0.232 | -0.076 | -0.228 | -0.289 | -0.421 | -0.257 | -0.251 |
| 17 | 1.799 | 2.604 | 1.895 | 0.091 | 2.092 | 2.039 | 1.753 | 17 | -0.209 | -0.071 | -0.205 | -0.251 | -0.386 | -0.236 | -0.226 |
| 16 | 1.775 | 2.570 | 1.867 | 0.085 | 2.092 | 2.021 | 1.735 | 16 | -0.190 | -0.066 | -0.187 | -0.229 | -0.357 | -0.219 | -0.208 |
| 15 | 1.738 | 2.529 | 1.838 | 0.078 | 2.067 | 1.989 | 1.707 | 15 | -0.177 | -0.064 | -0.177 | -0.209 | -0.338 | -0.210 | -0.196 |
| 14 | 1.693 | 2.489 | 1.802 | 0.073 | 2.015 | 1.950 | 1.670 | 14 | -0.174 | -0.064 | -0.174 | -0.200 | -0.330 | -0.207 | -0.192 |
| 13 | 1.636 | 2.442 | 1.748 | 0.069 | 1.926 | 1.887 | 1.618 | 13 | -0.177 | -0.067 | -0.178 | -0.192 | -0.335 | -0.216 | -0.194 |
| 12 | 1.559 | 2.403 | 1.686 | 0.063 | 1.802 | 1.814 | 1.555 | 12 | -0.191 | -0.077 | -0.195 | -0.181 | -0.358 | -0.239 | -0.207 |
| 11 | 1.446 | 2.365 | 1.591 | 0.058 | 1.622 | 1.694 | 1.463 | 11 | -0.230 | -0.103 | -0.244 | -0.172 | -0.408 | -0.292 | -0.242 |
| 10 | 1.196 | 2.310 | 1.359 | 0.050 | 1.285 | 1.442 | 1.274 | 10 | -0.328 | -0.188 | -0.374 | -0.158 | -0.512 | -0.428 | -0.331 |
| 9 | 0.573 | 1.891 | 0.699 | 0.041 | 0.625 | 0.774 | 0.767 | 9 | -0.402 | -0.568 | -0.560 | -0.139 | -0.525 | -0.627 | -0.470 |
| 8 | 0.292 | 1.124 | 0.334 | 0.037 | 0.363 | 0.413 | 0.427 | 8 | -0.291 | -0.787 | -0.428 | -0.132 | -0.397 | -0.516 | -0.425 |
| 7 | 0.234 | 0.715 | 0.250 | 0.036 | 0.305 | 0.324 | 0.311 | 7 | -0.237 | -0.684 | -0.347 | -0.128 | -0.336 | -0.448 | -0.363 |
| 6 | 0.217 | 0.553 | 0.224 | 0.035 | 0.292 | 0.295 | 0.269 | 6 | -0.208 | -0.575 | -0.304 | -0.126 | -0.300 | -0.407 | -0.320 |
| 5 | 0.213 | 0.516 | 0.218 | 0.035 | 0.290 | 0.284 | 0.259 | 5 | -0.193 | -0.524 | -0.279 | -0.126 | -0.278 | -0.384 | -0.297 |
| 4 | 0.217 | 0.503 | 0.223 | 0.036 | 0.308 | 0.289 | 0.263 | 4 | -0.188 | -0.506 | -0.271 | -0.127 | -0.270 | -0.371 | -0.289 |
| 3 | 0.227 | 0.495 | 0.235 | 0.042 | 0.334 | 0.295 | 0.271 | 3 | -0.199 | -0.507 | -0.280 | -0.147 | -0.280 | -0.374 | -0.298 |
| 2 | 0.316 | 0.695 | 0.321 | 0.082 | 0.390 | 0.429 | 0.372 | 2 | -0.387 | -0.804 | -0.482 | -0.306 | -0.525 | -0.658 | -0.527 |
| 1 | 0.941 | 1.373 | 1.021 | 0.080 | 1.183 | 1.116 | 0.952 | 1 | -0.446 | -0.760 | -0.602 | -0.181 | -0.609 | -0.701 | -0.550 |

Figure 6: The layer-wise alignment and uniformity analysis of GPT2-Large on six BEIR datasets. The **minimum** the loss, the **better** the alignment and uniformity. Conversely, the **maximum** the loss, the **worse** alignment and uniformity.

| #Layer | GPT2-XL (Alignment Loss) | | | | | | | #Layer | GPT2-XL (Uniformity Loss) | | | | | | |
|---|---|---|---|---|---|---|---|---|---|---|---|---|---|---|---|
| | COVID | NFCorpus | FiQA | ArguAna | SciFact | SCIDOCS | AVG | | COVID | NFCorpus | FiQA | ArguAna | SciFact | SCIDOCS | AVG |
| 49 | 0.786 | 1.169 | 0.784 | 0.354 | 0.586 | 0.933 | 0.769 | 49 | -1.285 | -1.616 | -1.649 | -1.545 | -1.465 | -1.480 | -1.507 |
| 48 | 0.131 | 0.341 | 0.160 | 0.050 | 0.129 | 0.231 | 0.174 | 48 | -0.182 | -0.409 | -0.265 | -0.183 | -0.276 | -0.299 | -0.269 |
| 47 | 0.162 | 0.634 | 0.188 | 0.055 | 0.165 | 0.279 | 0.247 | 47 | -0.209 | -0.543 | -0.313 | -0.203 | -0.339 | -0.362 | -0.332 |
| 46 | 0.238 | 1.095 | 0.266 | 0.064 | 0.254 | 0.403 | 0.387 | 46 | -0.268 | -0.641 | -0.415 | -0.238 | -0.461 | -0.493 | -0.419 |
| 45 | 0.328 | 1.393 | 0.372 | 0.071 | 0.370 | 0.545 | 0.513 | 45 | -0.323 | -0.553 | -0.496 | -0.267 | -0.552 | -0.567 | -0.460 |
| 44 | 0.410 | 1.545 | 0.463 | 0.077 | 0.477 | 0.651 | 0.604 | 44 | -0.352 | -0.461 | -0.530 | -0.289 | -0.601 | -0.579 | -0.469 |
| 43 | 0.493 | 1.651 | 0.554 | 0.082 | 0.578 | 0.754 | 0.685 | 43 | -0.371 | -0.391 | -0.539 | -0.307 | -0.621 | -0.574 | -0.467 |
| 42 | 0.570 | 1.728 | 0.637 | 0.087 | 0.665 | 0.848 | 0.756 | 42 | -0.386 | -0.344 | -0.544 | -0.330 | -0.634 | -0.564 | -0.467 |
| 41 | 0.642 | 1.795 | 0.716 | 0.094 | 0.742 | 0.925 | 0.819 | 41 | -0.394 | -0.308 | -0.541 | -0.347 | -0.638 | -0.548 | -0.463 |
| 40 | 0.714 | 1.853 | 0.798 | 0.099 | 0.817 | 1.000 | 0.880 | 40 | -0.400 | -0.280 | -0.538 | -0.367 | -0.641 | -0.534 | -0.460 |
| 39 | 0.775 | 1.911 | 0.865 | 0.105 | 0.884 | 1.073 | 0.936 | 39 | -0.404 | -0.260 | -0.534 | -0.386 | -0.643 | -0.527 | -0.459 |
| 38 | 0.847 | 1.967 | 0.939 | 0.109 | 0.958 | 1.143 | 0.994 | 38 | -0.405 | -0.242 | -0.526 | -0.399 | -0.643 | -0.517 | -0.455 |
| 37 | 0.927 | 2.022 | 1.020 | 0.115 | 1.041 | 1.220 | 1.058 | 37 | -0.408 | -0.221 | -0.513 | -0.418 | -0.643 | -0.503 | -0.451 |
| 36 | 0.992 | 2.070 | 1.087 | 0.120 | 1.111 | 1.285 | 1.111 | 36 | -0.409 | -0.206 | -0.503 | -0.429 | -0.639 | -0.486 | -0.445 |
| 35 | 1.073 | 2.128 | 1.170 | 0.122 | 1.195 | 1.364 | 1.175 | 35 | -0.401 | -0.189 | -0.482 | -0.428 | -0.625 | -0.461 | -0.431 |
| 34 | 1.140 | 2.176 | 1.242 | 0.125 | 1.271 | 1.434 | 1.231 | 34 | -0.392 | -0.174 | -0.462 | -0.436 | -0.610 | -0.439 | -0.419 |
| 33 | 1.199 | 2.227 | 1.307 | 0.129 | 1.345 | 1.492 | 1.283 | 33 | -0.382 | -0.163 | -0.447 | -0.434 | -0.598 | -0.425 | -0.408 |
| 32 | 1.280 | 2.279 | 1.386 | 0.129 | 1.429 | 1.561 | 1.344 | 32 | -0.366 | -0.150 | -0.425 | -0.420 | -0.577 | -0.402 | -0.390 |
| 31 | 1.341 | 2.326 | 1.451 | 0.132 | 1.490 | 1.620 | 1.393 | 31 | -0.360 | -0.139 | -0.409 | -0.418 | -0.572 | -0.388 | -0.381 |
| 30 | 1.405 | 2.363 | 1.523 | 0.135 | 1.576 | 1.681 | 1.447 | 30 | -0.340 | -0.128 | -0.381 | -0.416 | -0.547 | -0.364 | -0.363 |
| 29 | 1.469 | 2.393 | 1.584 | 0.132 | 1.663 | 1.742 | 1.497 | 29 | -0.313 | -0.115 | -0.346 | -0.402 | -0.514 | -0.334 | -0.337 |
| 28 | 1.525 | 2.420 | 1.633 | 0.131 | 1.751 | 1.795 | 1.543 | 28 | -0.289 | -0.105 | -0.319 | -0.391 | -0.487 | -0.313 | -0.317 |
| 27 | 1.567 | 2.441 | 1.671 | 0.128 | 1.821 | 1.837 | 1.578 | 27 | -0.269 | -0.096 | -0.291 | -0.371 | -0.462 | -0.289 | -0.296 |
| 26 | 1.600 | 2.458 | 1.699 | 0.124 | 1.875 | 1.864 | 1.603 | 26 | -0.249 | -0.088 | -0.267 | -0.349 | -0.433 | -0.266 | -0.275 |
| 25 | 1.629 | 2.474 | 1.720 | 0.119 | 1.922 | 1.888 | 1.625 | 25 | -0.235 | -0.082 | -0.248 | -0.328 | -0.416 | -0.252 | -0.260 |
| 24 | 1.659 | 2.476 | 1.737 | 0.114 | 1.976 | 1.912 | 1.646 | 24 | -0.217 | -0.075 | -0.227 | -0.312 | -0.390 | -0.233 | -0.242 |
| 23 | 1.681 | 2.485 | 1.745 | 0.106 | 2.030 | 1.932 | 1.663 | 23 | -0.198 | -0.069 | -0.205 | -0.284 | -0.364 | -0.216 | -0.223 |
| 22 | 1.693 | 2.480 | 1.748 | 0.101 | 2.067 | 1.939 | 1.671 | 22 | -0.183 | -0.064 | -0.187 | -0.262 | -0.341 | -0.201 | -0.206 |
| 21 | 1.691 | 2.467 | 1.743 | 0.096 | 2.084 | 1.935 | 1.669 | 21 | -0.170 | -0.060 | -0.173 | -0.246 | -0.321 | -0.187 | -0.193 |
| 20 | 1.688 | 2.451 | 1.734 | 0.090 | 2.102 | 1.930 | 1.666 | 20 | -0.155 | -0.055 | -0.156 | -0.224 | -0.298 | -0.172 | -0.177 |
| 19 | 1.680 | 2.434 | 1.724 | 0.086 | 2.105 | 1.919 | 1.658 | 19 | -0.146 | -0.052 | -0.146 | -0.209 | -0.281 | -0.163 | -0.166 |
| 18 | 1.668 | 2.418 | 1.717 | 0.081 | 2.096 | 1.903 | 1.647 | 18 | -0.140 | -0.050 | -0.140 | -0.197 | -0.269 | -0.156 | -0.159 |
| 17 | 1.654 | 2.411 | 1.712 | 0.076 | 2.070 | 1.892 | 1.636 | 17 | -0.138 | -0.049 | -0.138 | -0.184 | -0.264 | -0.155 | -0.155 |
| 16 | 1.640 | 2.411 | 1.703 | 0.071 | 2.053 | 1.890 | 1.628 | 16 | -0.138 | -0.050 | -0.139 | -0.172 | -0.264 | -0.156 | -0.153 |
| 15 | 1.601 | 2.392 | 1.679 | 0.064 | 1.978 | 1.853 | 1.595 | 15 | -0.143 | -0.054 | -0.145 | -0.156 | -0.271 | -0.164 | -0.156 |
| 14 | 1.548 | 2.370 | 1.642 | 0.060 | 1.882 | 1.797 | 1.550 | 14 | -0.156 | -0.060 | -0.159 | -0.150 | -0.286 | -0.180 | -0.165 |
| 13 | 1.485 | 2.352 | 1.591 | 0.054 | 1.769 | 1.729 | 1.497 | 13 | -0.173 | -0.072 | -0.180 | -0.140 | -0.312 | -0.205 | -0.180 |
| 12 | 1.381 | 2.334 | 1.506 | 0.049 | 1.588 | 1.617 | 1.413 | 12 | -0.208 | -0.096 | -0.222 | -0.133 | -0.359 | -0.253 | -0.212 |
| 11 | 1.196 | 2.269 | 1.343 | 0.044 | 1.317 | 1.423 | 1.265 | 11 | -0.258 | -0.128 | -0.291 | -0.128 | -0.415 | -0.329 | -0.261 |
| 10 | 0.894 | 2.107 | 1.065 | 0.039 | 0.949 | 1.111 | 1.028 | 10 | -0.315 | -0.251 | -0.389 | -0.124 | -0.457 | -0.435 | -0.329 |
| 9 | 0.394 | 1.554 | 0.502 | 0.033 | 0.429 | 0.537 | 0.575 | 9 | -0.284 | -0.561 | -0.420 | -0.112 | -0.379 | -0.473 | -0.372 |
| 8 | 0.155 | 0.479 | 0.177 | 0.030 | 0.181 | 0.212 | 0.206 | 8 | -0.177 | -0.492 | -0.255 | -0.105 | -0.248 | -0.316 | -0.266 |
| 7 | 0.130 | 0.340 | 0.145 | 0.029 | 0.153 | 0.180 | 0.163 | 7 | -0.157 | -0.394 | -0.219 | -0.101 | -0.222 | -0.283 | -0.229 |
| 6 | 0.119 | 0.293 | 0.129 | 0.027 | 0.140 | 0.163 | 0.145 | 6 | -0.144 | -0.343 | -0.197 | -0.099 | -0.205 | -0.261 | -0.208 |
| 5 | 0.111 | 0.263 | 0.117 | 0.026 | 0.129 | 0.153 | 0.133 | 5 | -0.138 | -0.322 | -0.183 | -0.098 | -0.193 | -0.252 | -0.198 |
| 4 | 0.108 | 0.251 | 0.114 | 0.026 | 0.127 | 0.151 | 0.130 | 4 | -0.139 | -0.321 | -0.182 | -0.097 | -0.191 | -0.248 | -0.196 |
| 3 | 0.118 | 0.279 | 0.125 | 0.031 | 0.141 | 0.169 | 0.144 | 3 | -0.161 | -0.387 | -0.215 | -0.114 | -0.225 | -0.291 | -0.232 |
| 2 | 0.169 | 0.388 | 0.166 | 0.047 | 0.191 | 0.235 | 0.199 | 2 | -0.272 | -0.596 | -0.316 | -0.175 | -0.396 | -0.462 | -0.370 |
| 1 | 0.962 | 1.620 | 1.089 | 0.179 | 1.038 | 1.232 | 1.020 | 1 | -0.842 | -1.527 | -1.093 | -0.470 | -1.144 | -1.274 | -1.058 |

Figure 7: The layer-wise alignment and uniformity analysis of GPT2-XL on six BEIR datasets. The **minimum** the loss, the **better** the alignment and uniformity. Conversely, the **maximum** the loss, the **worse** alignment and uniformity.

| #Layer | GPT-J-6B (Alignment Loss) | | | | | | | #Layer | GPT-J-6B (Uniformity Loss) | | | | | | |
|---|---|---|---|---|---|---|---|---|---|---|---|---|---|---|---|
| | COVID | NFCorpus | FiQA | ArguAna | SciFact | SCIDOCS | AVG | | COVID | NFCorpus | FiQA | ArguAna | SciFact | SCIDOCS | AVG |
| 29 | 0.541 | 0.982 | 0.476 | 0.229 | 0.486 | 0.709 | 0.571 | 29 | -0.880 | -1.667 | -1.161 | -0.961 | -1.520 | -1.583 | **-1.295** |
| 28 | 0.251 | 0.900 | 0.279 | 0.059 | 0.261 | 0.370 | 0.353 | 28 | -0.204 | -0.273 | -0.279 | -0.252 | -0.351 | -0.353 | -0.285 |
| 27 | 0.448 | 1.084 | 0.516 | 0.077 | 0.516 | 0.618 | 0.543 | 27 | -0.184 | -0.129 | -0.233 | -0.327 | -0.321 | -0.250 | -0.241 |
| 26 | 0.504 | 1.025 | 0.592 | 0.089 | 0.616 | 0.678 | 0.584 | 26 | -0.142 | -0.079 | -0.179 | -0.388 | -0.261 | -0.178 | -0.205 |
| 25 | 0.474 | 0.910 | 0.570 | 0.089 | 0.608 | 0.638 | 0.548 | 25 | -0.115 | -0.060 | -0.147 | -0.407 | -0.218 | -0.143 | -0.182 |
| 24 | 0.470 | 0.881 | 0.551 | 0.085 | 0.618 | 0.619 | 0.537 | 24 | -0.108 | -0.055 | -0.133 | -0.399 | -0.211 | -0.130 | -0.173 |
| 23 | 0.455 | 0.835 | 0.540 | 0.086 | 0.608 | 0.592 | 0.519 | 23 | -0.101 | -0.050 | -0.122 | -0.399 | -0.198 | -0.117 | -0.165 |
| 22 | 0.433 | 0.767 | 0.519 | 0.088 | 0.597 | 0.558 | 0.494 | 22 | -0.092 | -0.043 | -0.107 | -0.374 | -0.181 | -0.103 | -0.150 |
| 21 | 0.396 | 0.676 | 0.461 | 0.080 | 0.564 | 0.492 | 0.445 | 21 | -0.076 | -0.035 | -0.086 | -0.342 | -0.155 | -0.087 | -0.130 |
| 20 | 0.339 | 0.571 | 0.411 | 0.074 | 0.492 | 0.433 | 0.387 | 20 | -0.064 | -0.029 | -0.071 | -0.293 | -0.125 | -0.071 | -0.109 |
| 19 | 0.305 | 0.491 | 0.355 | 0.069 | 0.440 | 0.368 | 0.338 | 19 | -0.055 | -0.025 | -0.061 | -0.258 | -0.108 | -0.061 | -0.095 |
| 18 | 0.266 | 0.426 | 0.314 | 0.069 | 0.396 | 0.320 | 0.299 | 18 | -0.048 | -0.022 | -0.053 | -0.257 | -0.095 | -0.053 | -0.088 |
| 17 | 0.228 | 0.359 | 0.266 | 0.068 | 0.350 | 0.283 | 0.259 | 17 | -0.041 | -0.018 | -0.044 | -0.237 | -0.077 | -0.047 | -0.077 |
| 16 | 0.192 | 0.300 | 0.213 | 0.055 | 0.293 | 0.229 | 0.214 | 16 | -0.035 | -0.016 | -0.038 | -0.191 | -0.066 | -0.041 | -0.065 |
| 15 | 0.155 | 0.232 | 0.171 | 0.050 | 0.230 | 0.186 | 0.171 | 15 | -0.030 | -0.014 | -0.032 | -0.173 | -0.054 | -0.035 | -0.056 |
| 14 | 0.125 | 0.181 | 0.132 | 0.046 | 0.185 | 0.144 | 0.136 | 14 | -0.025 | -0.012 | -0.027 | -0.156 | -0.045 | -0.030 | -0.049 |
| 13 | 0.104 | 0.151 | 0.110 | 0.041 | 0.158 | 0.114 | 0.113 | 13 | -0.022 | -0.010 | -0.023 | -0.135 | -0.039 | -0.025 | -0.042 |
| 12 | 0.086 | 0.129 | 0.088 | 0.032 | 0.137 | 0.094 | 0.094 | 12 | -0.019 | -0.009 | -0.020 | -0.102 | -0.034 | -0.022 | -0.034 |
| 11 | 0.089 | 0.134 | 0.089 | 0.029 | 0.149 | 0.098 | 0.098 | 11 | -0.017 | -0.008 | -0.018 | -0.091 | -0.031 | -0.020 | -0.031 |
| 10 | 0.114 | 0.178 | 0.115 | 0.027 | 0.206 | 0.128 | 0.128 | 10 | -0.016 | -0.008 | -0.017 | -0.076 | -0.030 | -0.019 | -0.028 |
| 9 | 0.136 | 0.211 | 0.136 | 0.025 | 0.254 | 0.154 | 0.153 | 9 | -0.015 | -0.007 | -0.016 | -0.072 | -0.029 | -0.018 | -0.026 |
| 8 | 0.159 | 0.249 | 0.159 | 0.024 | 0.299 | 0.180 | 0.178 | 8 | -0.015 | -0.007 | -0.015 | -0.070 | -0.029 | -0.017 | -0.026 |
| 7 | 0.170 | 0.264 | 0.165 | 0.021 | 0.337 | 0.191 | 0.191 | 7 | -0.013 | -0.006 | -0.013 | -0.057 | -0.026 | -0.016 | -0.022 |
| 6 | 0.197 | 0.305 | 0.192 | 0.019 | 0.387 | 0.223 | 0.221 | 6 | -0.012 | -0.006 | -0.012 | -0.050 | -0.024 | -0.015 | -0.020 |
| 5 | 0.188 | 0.285 | 0.182 | 0.018 | 0.395 | 0.212 | 0.213 | 5 | -0.010 | -0.005 | -0.010 | -0.051 | -0.020 | -0.012 | -0.018 |
| 4 | 0.201 | 0.306 | 0.195 | 0.016 | 0.419 | 0.227 | 0.227 | 4 | -0.009 | -0.005 | -0.009 | -0.039 | -0.019 | -0.012 | -0.016 |
| 3 | 0.072 | 0.226 | 0.085 | 0.018 | 0.065 | 0.105 | 0.095 | 3 | -0.136 | -0.357 | -0.196 | -0.054 | -0.177 | -0.233 | -0.192 |
| 2 | 0.049 | 0.152 | 0.058 | 0.014 | 0.043 | 0.071 | 0.065 | 2 | -0.100 | -0.276 | -0.147 | -0.045 | -0.128 | -0.175 | -0.145 |
| 1 | 0.013 | 0.043 | 0.015 | 0.003 | 0.011 | 0.019 | **0.017** | 1 | -0.034 | -0.109 | -0.050 | -0.009 | -0.041 | -0.060 | -0.051 |

Figure 8: The layer-wise alignment and uniformity analysis of GPT-j-6B on six BEIR datasets. The **minimum** the loss, the **better** the alignment and uniformity. Conversely, the **maximum** the loss, the **worse** alignment and uniformity.

The analysis code for estimating the layer-wise alignment loss and uniformity loss is available in the supplementary material.

