# OpenReview forum: "LLM-Oriented Retrieval Tuner"
_ICLR.cc/2024/Conference — Submitted to ICLR 2024_

### Official Review · Reviewer_VDA7 · 2023-10-23

**Soundness:** 3 good
**Presentation:** 2 fair
**Contribution:** 2 fair
**Rating:** 3
**Confidence:** 4

**Summary:**

This paper proposes a method to adapt a frozen LLM for dense retrieval, by training a shallow (3-layer) adaptor transformer. In particular, for a given LLM, they employ the alignment and uniformity analysis (Wang and Isola, 2020) to find the layers with the best "alignment" and "uniformity", which become the input of the adaptor transformer. The adaptor transformer then leverages self- and cross-attention to fuse the hidden representations from these two layers, and produces a single dense vector as its final output for retrieval.

The proposed LMORT method is evaluated on 6 BEIR datasets (which were also used for the alignment and uniformity analysis). The authors show that LMORT works with different LLMs, up to GPT-j-6B, outperforming baseline using a single layer (e.g. best alignment layer, best uniformity layer, or embedding layer). When using GPT-j-6B, it also outperforms baseline DRs such as BM25 and DPR (although it misses comparison with, and performs worse than, more recent DRs in the literature such as Contriever and DRAGON). It also performs worse than cpt-text-L (GPT3-6B) in general, but cpt-text-L finetunes the entire LLM while LMORT trains a shallow adaptor while freezing the LLM.

**Strengths:**

- The idea of adapting a frozen LLM for dense retrieval using a shallow transformer is interesting, and leveraging multiple layers from the LLM based on alignment and uniformity analysis is a sound approach.

- Experiments show that the proposed approach indeed works with multiple LLMs of various sizes, and outperforms simpler baseline approaches that utilize a single layer.

**Weaknesses:**

- The same 6 BEIR datasets used for evaluation are also used in the alignment and uniformity analysis, which violates the zero-shot setting of BEIR and makes the comparison unfair with the other retrievers. The authors should at least evaluate on additional datasets that were not seen during the analysis. Otherwise, the gain from this alignment and uniformity analysis may have been attained from fitting to the test set.

- The paper does not compare with more recent DRs, which achieve comparable or much higher accuracy than the proposed method despite using much smaller encoders. For example, Contriever-MSMARCO (https://arxiv.org/abs/2112.09118) achieves an average of 0.424 NDCG@10 on the 6 BEIR tasks, whereas the more recent DRAGON model (https://arxiv.org/abs/2302.07452) achieves 0.460 on average. Both models are BERT-base models, which is 50x smaller than the GPT-j-6B used in this work, yet DRAGON outperforms LMORT by 3.5 points on average in NDCG.

- While the proposed method is interesting, I find it difficult to justify its practical use. If it is intended to use as a standalone dense retriever, it performs significantly worse than many recent DRs while using a 50x larger encoder, which makes the embeddings much larger (due to higher dimensionality) and latency much higher.
An unexplored alternative application would have been more appealing, which is retrieval-augmented generation (RAG). By adapting a frozen LLM for retrieval, it would indeed be cheaper to encode the queries and can be applied to methods such as kNN-LM (https://arxiv.org/abs/1911.00172). Unfortunately, this direction is not examined in this paper.

**Questions:**

- Table 2 compares only with LMORT variants that use a single layer (or two) of the LLM. Have you compared with additional baselines of fusing more layers of the LLMs (either via attention similar to that in LMORT, or via simpler pooling mechanisms)?
Have you also tried layer selecting methods not based on alignment or uniformity? For instance, choosing first and last layer?

- Why do you only evaluate on 6 datasets from BEIR out of 18? It is unclear to me why the rest of BEIR is omitted and makes the results less convincing.

---

> ### Author Response · Authors · 2023-11-21
>
> Thank you for your review. There appears to be some differences between your comment and our work goals. We would appreciate your review of the explanation provided in our **General Response**.
>
> **1. About Zero-Shot Setting**
>
> We respectfully disagree with your claims about violating zero-shot evaluation and fitting the test set. Our retrieval module neither uses any evaluation dataset for fine-tuning nor testing data to select model checkpoints. The alignment and uniformity analysis is also not to evaluate the retrieval performance of each LLM layer on the evaluation datasets.
> Furthermore, we have supplemented the alignment and uniformity analysis for Train and Dev sets in these evaluation datasets (noting that not all datasets include both). Analysis results consistently show stable rankings for each LLM layer across Train, Dev, and Test sets. These results underscore that the alignment and uniformity analysis in our paper doesn't fit the test set.
>
> You can find the analysis results through these anonymous links: [[GPT2-Large]](https://drive.google.com/file/d/1hr-AL-SqPSKEj9HkD4_9dTfXKL5l2tXe/view?usp=sharing), [[GPT2-XL]](https://drive.google.com/file/d/1cmAennw-yalv7zkJdG8uVEDA9V8mEHT_/view?usp=sharing), [[GPT-j-6B]](https://drive.google.com/file/d/1g0KLqvJZOB9orXvrVXNRdHDwRr5BVUnS/view?usp=sharing).
>
>
> **2. About Questions**
>
> The paper has explained the reasons for using the six BEIR datasets for evaluation (P4)--conducting inference on a larger corpus with LLMs surpassed our current computational and storage resources.
>
> Additionally, our current method design proved to be the most effective among the various solutions we  explored, including the options you mentioned such as simple pooling, fusing additional LLM layers, or using the first and last LLM layers directly.

---

> > ### Comment · Reviewer_VDA7 · 2023-11-22
> >
> > After reading the authors' response, I'd like to maintain my original assessment.
> >
> > - About Contributions
> > The response claims that the paper's main focus is to "enhance the long-term memory capabilities of generative LLMs". However, no experiment is conducted in this setting, where the retrieval index would need to be built from the LLM representations of past contexts. As I mentioned in the original review, this direction (which is more similar to kNN-LM) would indeed have made more sense than using the LLM as a standalone retriever, unfortunately the paper provides no empirical experiments in this setting.
> >
> > - About Zero-Shot
> > In the response, the authors emphasized that the analysis results are similar across the train, dev, and test splits of each evaluation dataset. This, unfortunately, does not address my concern. The alignment analysis in this paper requires access to labeled data, which makes the comparison with existing results on BEIR unfair. To properly compare with retrieval results on BEIR, no labeled data should be used from the target datasets at all. Instead, out-of-domain labeled data such as MS MARCO can be used, as proposed in the original BEIR paper. If the authors can show that their analysis can be done on MS MARCO, and generalize well to the 6 BEIR datasets, it would have been much more convincing.

---

> ### Author Response · Authors · 2023-11-23
>
> **1. About Zero-Shot Setting**
>
> Thank you sincerely for your insightful suggestion. In response, we have augmented the alignment analysis results for the MS MARCO dataset. Considering the large scale of the complete MARCO dataset, our alignment analysis was conducted specifically on the first 5,000 training queries and their corresponding positive passages.
>
> [[GPT2-Large-Marco-Alignment]](https://drive.google.com/file/d/1rPfhFKDX2L_-I8kmiDtcAIuLVpqKtQAg/view?usp=sharing)
>
> [[GPT2-XL-Marco-Alignment]](https://drive.google.com/file/d/1Xmp0tZogFSXQWNzajmi7WxDnPe_2JTO7/view?usp=sharing)
>
> [[GPT-j-6B-Marco-Alignment]](https://drive.google.com/file/d/19pm-vNdQOShMf5O_r22QebSoDHXMhvOa/view?usp=drive_link)
>
> The analysis results above demonstrate that the optimal alignment layers on GPT-2-large, GPT-2-XL, and GPT-j-6B are consistent with the alignment layers selected in our paper. We hope these supplemental analysis results address your concerns about our zero-shot setups.
>
> **1. About KNN-LM**
>
> In contrast to our vision, KNN-LM stores memory at the token level. During memory retrieval, KNN-LM calculates similarities using current query information and all memory tokens. However, we envision long-term memory to be hierarchical. When LLM accesses long-term memory, it should initially use current query information to retrieve the sequence-level memory granularity.
>
> We think it to be an important question enough to investigate how LLM can be empowered to store and retrieve sequence-level memory (text retrieval) while maintaining compatibility with text processing and generation capabilities.
>
> But, we genuinely agree with your viewpoint that a thorough verification of the entire process of memory storage, retrieval, and utilization would contribute to a clearer and more comprehensive understanding of this work. We will continue to explore to attain this objective.

---

### Official Review · Reviewer_Zx7X · 2023-10-30

**Soundness:** 2 fair
**Presentation:** 3 good
**Contribution:** 3 good
**Rating:** 5
**Confidence:** 4

**Summary:**

The paper presents an attempt to merge Dense Retrieval (DR) with Large Language Models (LLM) like GPT-3 and GPT-4, aiming to enhance LLM's memory capacity. At its core, the challenge remains the disconnect between LLM text generation and DR. The authors introduce the LLM-Oriented Retrieval Tuner (LMORT) as a solution, suggesting it can integrate LLM and DR without altering the LLM. Through their analysis, they emphasize the tension between alignment and uniformity within LLM layers, implying that achieving one often comes at the expense of the other.

Their proposed LMORT method, built on a Transformer-like structure, claims to strike a balance between alignment and uniformity by synergizing the best features of LLM's representation space. Extensive tests on six BEIR datasets are presented as evidence, with results indicating that LMORT's zero-shot retrieval performance improves by 13% as the LLM size increases. However, when compared against robust DR models, LMORT's advantages seem less pronounced, only slightly edging out competitors with minimal layer tuning. While the authors highlight LMORT's efficiency in harnessing LLM's strengths and mitigating their incompatibilities, the presented reductions in training parameters and time raise questions about possible trade-offs. The paper, while presenting an intriguing approach, requires a more comprehensive exploration and validation to truly ascertain LMORT’s potential in the DR landscape.

**Strengths:**

1. LMORT's bidirectional attention sub-layers are an interesting twist on the usual Transformer setup. It's evident they're trying something new by pulling in features from LLM’s alignment and uniformity layers. The idea of mixing and matching these layers is creative, though I remain unconvinced about how revolutionary this really is.

2. One thing I've got to appreciate about LMORT is its aim for a leaner model. They've tried to cut back on unnecessary layers, which could mean it's efficient. The dense vector output is a clear nod towards streamlining, but whether it's a game-changer for dense retrieval is a question I still have.

3. LMORT is trying to bridge the existing capabilities of LLMs for dense retrieval tasks. It's commendable that they’re looking for synergy and not attempting to reinvent everything from scratch. But again, whether it truly harmonizes the best of both worlds is something I believe time (and rigorous testing) will reveal.

**Weaknesses:**

1. The objective function of LMORT, given by:
$$
\theta^* = \text{argmin}_{\theta} \sum_{X_q \in Q} \sum_{X^+ \in C^+_{X_q}} -\log \left( \frac{ \exp(\text{sim}(X_q, X^+; \theta, \tilde{\phi}_{llm}))}{\exp(\text{sim}(X_q, X^+; \theta, \tilde{\phi}_{llm})) + \sum_{X^-_p \in C^-_{X_q}} \exp(\text{sim}(X_q, X^-_p; \theta, \tilde{\phi}_{llm}))} \right)
$$
seems to treat all negative pairs equally, pushing their similarity scores toward 0. This could result in a lack of nuanced understanding of varying degrees of negativity or irrelevance among samples. Consequently, as `top_k` retrieval results increase, this undifferentiated approach may adversely affect the precision and relevance of retrieval results, leading to poorer model performance in larger retrieval tasks.


2. The method relies heavily on the frozen LLM, which might constrain the adaptability of the LMORT in diverse real-world retrieval scenarios where the underlying pretrained model isn't perfectly aligned with the task at hand.

3. The paper doesn't seem to discuss or account for potential edge cases where the bifurcation between alignment and uniformity layers might cause misrepresentations or omissions of critical data.

**Questions:**

**Questions:**

1. Within the objective function, there's a pronounced use of the negative logarithm operating on an exponential function related to similarity measures. This raises concerns:
\[
\theta^* = \text{argmin}_{\theta} \sum_{X_q \in Q} \sum_{X^+ \in C^+_{X_q}} -\log \left( \frac{ \exp(\text{sim}(X_q, X^+; \theta, \tilde{\phi}_{llm}))}{\exp(\text{sim}(X_q, X^+; \theta, \tilde{\phi}_{llm})) + \sum_{X^-_p \in C^-_{X_q}} \exp(\text{sim}(X_q, X^-_p; \theta, \tilde{\phi}_{llm}))} \right)
\]
   Could this cause potential issues in terms of stability or convergence, especially with extremely high or low similarity scores?

2. Is there a comparative analysis with other retrieval methods in terms of computational resources and time? If not, wouldn't this be essential to truly claim the efficiency of the proposed method?

---

> ### Author Response · Authors · 2023-11-21
>
> We genuinely appreciate your insightful comments on our work, encompassing both summaries and concerns. Moreover, certain perspectives you've shared have been inspiring to us. We have explained your concerns regarding the computational cost in our **General Response** for your consideration.
>
> **1. About Training Objective**
>
> We adopt a widely-used cross-entropy loss in DR model training, such as [GTR](https://aclanthology.org/2022.emnlp-main.669.pdf) and [CPT](https://arxiv.org/pdf/2201.10005.pdf), without introducing any modifications. We would like to provide a gentle reminder that the concerns you've raised regarding this loss function appear to extend beyond the scope of our current work. However, in light of your concerns, we have examined the curves of the model training losses. Our experimental results suggest that the stability and convergence of the training objective is robust. You can find the training curves in [[this anonymous link]](https://drive.google.com/file/d/1FWOuEQrG9cvJWuAYP0Q21QJvmir7-0bA/view?usp=drive_link).
>
> **2. About Frozen LLM**
>
> Our retrieval module is specifically tailored for a frozen LLM, denoting an LLM that has not undergone fine-tuning for a DR task. Theoretically, our method can be applied to LLMs that have been fine-tuned for various text processing and generation tasks. We are conducting experiments on [LLAMA2-7B-Chat](https://huggingface.co/meta-llama/Llama-2-7b-chat-hf), an LLM fine-tuned on diverse instruction datasets. Preliminary results suggest that this LLM displays a consistent mutually exclusive phenomenon of uniformity and alignment across different layers, in line with our paper's findings. You can find the analysis results in [[this anonymous link]](https://drive.google.com/file/d/1Qel8mllDdV-GDolHvJfQLVHkWjVpZcY6/view?usp=sharing).
>
> **3.  About Edge Cases**
>
> We would sincerely appreciate it if you could provide additional details or further elaborate on the edge cases mentioned in "Weakness 3".

---

### Official Review · Reviewer_CcZQ · 2023-11-05

**Soundness:** 2 fair
**Presentation:** 3 good
**Contribution:** 3 good
**Rating:** 5
**Confidence:** 3

**Summary:**

Due to the paradigm discrepancy between text generation of LLM and DR, it is still an open challenge to integrate the retrieval and generation tasks in a shared LLM.

The paper proposes an efficient LLM-Oriented Retrieval Tuner (LMORT), which decouples DR capacity from base LLM and non-invasively coordinates the optimally aligned and uniform layers of the LLM towards a unified DR space, achieving an efficient and effective DR without tuning the LLM itself. The proposed method could achieve competitive zero-shot retrieval performance compared to a range of strong DR models while maintaining the generation ability of LLM.

**Strengths:**

(1) The studied problem is an emerging topic in LLMs.

(2) The proposed method has good intuitions.

(3) The paper is well presented and written.

**Weaknesses:**

(1) The paper may further explore the capability of the proposed method, when using other (larger) LLMs, such as Llama 2 and falcon (or Flan-T5). The behavior of different LLMs may vary. Although the paper claims general contributions to 'LLM-Oriented Retrieval Tuner', limited LLMs are used in the evaluation.

(2) The paper may further discuss the additional latency by the proposed approach. When using a larger model, such as Llama 2, will the proposed method's retrieval and inference latency/cost be much higher, compared to some baselines?

(3) The designed approach seems to rely on the assumption that the LLM models can be accessed. Currently, there are very powerful LLMs which only provides API, where the proposed method can not be directly applied?

Considering (2) and (3), for the practical use purpose, it might be unclear whether 'integrate the retrieval and generation tasks in a shared LLM' is an prominent direction to explore in general.

**Questions:**

(1) Can the proposed work in some more recent LLMs (e.g., Llama 2 or falcon)? Although the paper claims general contributions to 'LLM-Oriented Retrieval Tuner', limited LLMs are used in the evaluation.

(2) When using a larger model, such as Llama 2, will the proposed method's retrieval and inference latency/cost be much higher, compared to some baselines?

---

> ### Author Response · Authors · 2023-11-21
>
> Thank you for your review and suggestions. We have explained some of your concerns in the **General Response** for your consideration.
>
> **1. About Larger LLM**
>
> We are currently conducting experiments with LLAMA2-7B and have observed a consistent manifestation of layered exclusion in alignment and uniformity. These findings align with our previous observations in the GPT series LLM. Please see [[this anonymous link]](https://drive.google.com/file/d/1K4A-8FYlyQh7k_CAVbNZETEhJrbfzLdT/view?usp=sharing) for the result.
>
> **2. About Closed-Source LLM**
>
> In the case of closed-source LLMs, our method is applicable if their API provides access to the hidden states of the optimal alignment layer and the uniform layer, eliminating the need to open-source the parameters of the LLM.
> Additionally, we would like to gently convey the perspective that the worth of research should not be exclusively defined by closed-source APIs.

---

### Official Review · Reviewer_XYMY · 2023-11-08

**Soundness:** 3 good
**Presentation:** 2 fair
**Contribution:** 2 fair
**Rating:** 5
**Confidence:** 3

**Summary:**

This paper introduces a new method, LMORT, to adapt large language models (e.g., GPT-2 Large, GPT-2 XL, GPT-j-6B) for retrieval tasks, by inserting new LMORT layers taking intermediate representations of two layers in the original LLM, which are selected based on their uniform loss and alignment loss. The experimental method shows that their final model outperforms their four ablated LMORT versions, while still largely lagging behind other methods, including BM25 and DPR.
I have several major concerns about this paper. First, for me the motivations (unifying alignment and uniformity of LLM representation spaces) itself as well as the connection between the motivation and the proposed method is unclear. Also, experimental results are weak and lack baselines, which do not provide convincing results to support the proposed method.
While it is not a major concern, there are several minor issues in writing (e.g., citation formats, abbreviation, introduction). I discussed this in detail in weaknesses. Overall, I don't recommend acceptance in the current form.

**Strengths:**

- The paper is easy to follow
- Interesting observations about different layer repersentaitons of LLMs in text retrieval tasks.

**Weaknesses:**

1. **Unclear motivations and lack of the connections between the motivations and the proposed method design**: I understand that alignment matters in text retrieval but why does uniformity matter? While the authors cited Wang and Isola (2020) briefly, papers should be self-containing and as the cited prior work focuses on non-text domains, more careful consideration and discussions on whether this assumption holds in the text retrieval task should be discussed more. if the final goal based on this motivation is to make the two properties distributed across different layers, why adding the proposed layers can be an effective solutions? Are there any alternative solutions? For instance, several work introduces a new objective to align different layers in transformers. I think this proposed method could be one solution, but explanations on why this is the best way to the discussed issue should be included.

2. **Weak experimental results and lack of baselines**:
The proposed method largely lags behind other retrieval models, including BM25 or DPR. Moreover, methods such as Contriever-MSMARCO (similar size as DPR, bert-base) have shown even much better performance than the listed baselines. While the authors claim this proposed method only added a few layers to be updated, how many parameters are exactly updated during training? Even with full parameter updates, if the base model is BERT-base, still the number of the trainable parameters can be less than the proposed method based on 6B LMs. Moreover, billions of parameters (both frozen and trained) are used during inference, which can significantly increase the inference costs. Also, what happens if we simply insert adapters and only update adapter layers during training?

**Questions:**

- Have you compared the proposed method with a baseline which simply inserts adapter layers to GPT2 or GPT--6B?

Minor typos & writing suggestions
- According to the ICLR template guide, unless the citation is a part of the text (e.g., X et al. (2028) found), the citations should be in parenthesis using \verb|\citep{}| (as in ``Deep learning shows promise to make progress
towards AI~\citep{Bengio+chapter2007}.''). Many citations in this paper, particularly Section 2 and Section 3 dropped parenthesis.
- I found "larger LLMs" sounds strange, as LLMs themselves stand as "large language models", What is the boundary between larger large language models and smaller large language models?
- Several abbreviations are defined multiple times throughout the papers e.g., DR is defined twice in the introduction. You don't need to define the same abbreviations multiple times.

---

> ### Author Response · Authors · 2023-11-21
>
> Thank you for your review. Our work's objectives may have some distinctions from your interpretation. We would appreciate it if you could review our explanation provided in the **General Response**.
>
> **1. About Uniformity**
>
> Many studies have demonstrated the significance of uniformity in text representation learning tasks, such as  [SimCSE](https://arxiv.org/pdf/2104.08821.pdf), [COCO-DR](https://arxiv.org/pdf/2210.15212.pdf), and [this work](https://openreview.net/pdf?id=MxvHVNukama).
>
> Furthermore, a prototypical counterexample highlights the important role of uniformity in text representation learning. In a collapsed representation space where all text is projected to the same representation vector, alignment is good but uniformity is lacking, making such spaces unsuitable for retrieval.
>
> **2. About Writing**
>
> Thank you very much for your suggestions regarding some citation formats and terminology descriptions. We have already revised these parts.

---

> > ### Comment · Reviewer_XYMY · 2023-11-30
> > **Thanks for your response.**
> >
> > I checked both individual and general response and I Increased my score to 5.

---

### Author Response · Authors · 2023-11-21
**General Response**

Many thanks to all the reviewers. We would like to offer a more detailed explanation of the following three points.

**1. About Contributions**

This work does not aim to adapt LLMs for retrieval tasks (dense retrieval, DR) or fine-tune a retriever to fetch external knowledge for enhancing LLMs (retrieval-augmented LLM).  Instead, it contributes to a new fancy direction to enhance the long-term memory capabilities of generative LLMs.

Specifically, the new direction is to empower the generative LLM (potentially severed as an AGI Agent) by converting processed or generated information's hidden states directly into dense vectors, stored as memories.  When handling new tasks, the LLM transforms the hidden states of current input information into query vectors, retrieving relevant hidden states from its memory, akin to the human brain. This process enhances the LLM's task execution without redundantly encoding the processed raw textual information using prompt formats.

To achieve the above fancy vision, this paper explores a key question in the emerging research path: Can a generative LLM be compatible with text retrieval functions? To investigate this, we adopt the setting of a frozen generative LLM because any DR-task fine-tuning, even with parameter-efficient tuning methods, hinders the repurposing of hidden states generated during the forward process for other text processing and generation tasks. This challenge arises from the discrepancy between the retrieval and generation task paradigms.

Our analysis revealed that directly using frozen LLM's hidden states doesn't create an efficient retrieval space. However, hidden states from different layers possess alignment and uniformity, which are crucial for effective retrieval. Building on these findings, we introduce a retrieval module based on LLM's hidden states. This module is designed to store and retrieve LLM-processed information by merging text representations from the LLM's optimal alignment and uniformity layers into a unified output space. It can integrate into the standard forward process of a generative LLM, eliminating the necessity to re-encode the raw text of processed information.

Experimental results from six zero-shot retrieval datasets demonstrate a 13% performance enhancement in the retrieval module as the LLM scales from 0.75B to 6B, highlighting its scalability. Moreover, the retrieval module maintains efficiency in parameter scale relative to the overall LLM. With the 6B LLM, the retrieval module's parameter number reduces to 0.13B after dimensionality reduction, indicating that activating LLM's information retrieval function comes at a mere 2% additional parameter cost.

**2. About Baselines**

Our main baseline is the retriever built on the hidden states of the frozen LLM, with experimentation involving four similar baselines. Traditional DR methods follow a different technical approach from ours. Nevertheless, for reference, we included three dense retrieval methods in our experiments, employing simlilar training data and strategy.
The recent SOTA DR methods (mentioned by Reviewers XYMY and VDA7) improve performance through techniques such as DR-oriented pre-training, knowledge distillation, and data augmentation—approaches parallel to ours. These methods can similarly enhance the performance of our approach. Additionally, adapter tuning the LLM as a retriever by inserting the adapter module into each LLM layer raises compatibility issues between LLM's functions for both text generation and retrieval, deviating from our original intent. Therefore, we think that comparison with these baselines is not necessary.

**3. About Inference Cost**

Our module empowers LLM to directly leverage its already generated hidden states for storing and retrieving processed historical information. Therefore, the inference cost of our retrieval module is confined to its own computation, excluding the LLM.
We show the parameter number and average retrieval performance results of our retrieval module (LMORT) using the frozen GPT-j-6B. Additionally, we compare these results with two traditional DR models, where inference cost and parameter number exhibit a proportional relationship.

|Method|#Parameters|Avg NDCG@10|
|:---|:---|:---|
|DPR|0.11B|0.339|
|GTR-XXL|4.5B|0.445|
|LMORT|0.8B|0.425|
|LMORT (dimension-reduced)|0.13B|0.412|

---

### Meta-Review · Area_Chair_ZE1w · 2023-12-06

**Metareview:**

This paper introduces LMORT, a method to adapt large language models (e.g., GPT-2 Large, GPT-2 XL, GPT-j-6B) for retrieval tasks. It achieves this by incorporating new LMORT layers, which utilize intermediate representations from two selected layers in the original LLM. These layers are chosen based on their uniform loss and alignment loss.

The paper is coherent and clear. It addresses a practical and significant problem in the field. The experiments demonstrate that the proposed approach is effective across various sizes of LLMs, surpassing simpler baseline methods that rely on a single layer.

However, reviewers (XYMY & VDA7) express reservations about the technical contributions of the submission and its comparisons with baselines. This is in light of strong retrieval models that operate without supervised data, such as DRAGON and Contriever.

While the authors have provided further details on zero-shot retrieval evaluation and the significance of their work, I believe the paper could be strengthened by more emphatically addressing these two areas. This would more convincingly demonstrate LMORT's effectiveness. Therefore, I recommend a reject rating.

**Justification For Why Not Higher Score:**

As previously noted, reviewers (XYMY & VDA7) have expressed concerns regarding the technical contributions of the submission and its baseline comparisons. This perspective arises particularly when considering strong retrieval models like DRAGON and Contriever, which function without supervised data. Although the authors have further elaborated on zero-shot retrieval evaluation and the importance of their work, I suggest that the paper would benefit from a more assertive approach in addressing these two specific areas.

**Justification For Why Not Lower Score:**

N/A

---

### Decision · Program_Chairs · 2024-01-16

Reject